

# Nitrogen eutrophication particularly promotes turf algae in coral reefs of the central Red Sea

Denis B. Karcher[1], Florian Roth[2,3,4], Susana Carvalho[2], Yusuf C. El-Khaled[1], Arjen Tilstra[1], Benjamin Kürten[2,5], Ulrich Struck[6,7], Burton H. Jones[2] and Christian Wild[1]

[1] Marine Ecology Department, Faculty of Biology and Chemistry, University of Bremen, Bremen, Germany
[2] Red Sea Research Center, King Abdullah University of Science and Technology (KAUST), Thuwal, Saudi Arabia
[3] Baltic Sea Centre, Stockholm University, Stockholm, Sweden
[4] Tvärminne Zoological Station, Faculty of Biological and Environmental Sciences, University of Helsinki, Helsinki, Finland
[5] Project Management Jülich, Jülich Research Centre, Rostock, Germany
[6] Museum für Naturkunde, Leibniz Institute for Evolution and Biodiversity Science, Berlin, Germany
[7] Department of Earth Sciences, Freie Universität Berlin, Berlin, Germany

Corresponding author
Denis B. Karcher,
db.karcher@gmx.de

## ABSTRACT

While various sources increasingly release nutrients to the Red Sea, knowledge about their effects on benthic coral reef communities is scarce. Here, we provide the first comparative assessment of the response of all major benthic groups (hard and soft corals, turf algae and reef sands—together accounting for 80% of the benthic reef community) to in-situ eutrophication in a central Red Sea coral reef. For 8 weeks, dissolved inorganic nitrogen (DIN) concentrations were experimentally increased 3-fold above environmental background concentrations around natural benthic reef communities using a slow release fertilizer with 15% total nitrogen (N) content. We investigated which major functional groups took up the available N, and how this changed organic carbon ($C_{org}$) and N contents using elemental and stable isotope measurements. Findings revealed that hard corals (in their tissue), soft corals and turf algae incorporated fertilizer N as indicated by significant increases in $\delta^{15}N$ by 8%, 27% and 28%, respectively. Among the investigated groups, $C_{org}$ content significantly increased in sediments (+24%) and in turf algae (+33%). Altogether, this suggests that among the benthic organisms only turf algae were limited by N availability and thus benefited most from N addition. Thereby, based on higher $C_{org}$ content, turf algae potentially gained competitive advantage over, for example, hard corals. Local management should, thus, particularly address DIN eutrophication by coastal development and consider the role of turf algae as potential bioindicator for eutrophication.

# INTRODUCTION

Coral reefs are among the most productive and biologically diverse ecosystems on the planet (*Roberts, 2002*), even though they grow in oligotrophic waters of the tropics (*Odum & Odum, 1955*). The young and isolated Red Sea, with its thriving coral reefs, is highly oligotrophic, particularly in the subtropical central and northern areas (*Raitsos et al., 2013*; *Sawall et al., 2014b*; *Kürten et al., 2014*; *Roth et al., 2018*). However, nutrient inputs to the Red Sea from aquaculture (*Loya et al., 2004*; *Kürten et al., 2015*; *Dunne, 2018*; *Hozumi et al., 2018*) and urban waste water (*Basaham et al., 2009*; *Al-Farawati, 2010*; *Kürten et al., 2014*; *Peña-García et al., 2014*) affect marine life (*Loya et al., 2004*; *Mohamed & Mesaad, 2007*; *Basaham et al., 2009*; *Naumann et al., 2015*). At the same time, the expansion of aquaculture industries in view of the Saudi Arabian coastal development agenda (https://vision2030.gov.sa/en/node), and growing urban sources, for example, from the city of Jeddah with about 4.6 Mio. inhabitants (*Ministry of Health, 2017*), represent further stressors to coral reefs in the Red Sea. Significant parts of the city rely on septic tanks for wastewater which can be a source of nutrients and pollutants through leakages into the groundwater (*Abu-Rizaiza & Sarikaya, 1993*; *Aljoufie & Tiwari, 2015*). Moreover, the discharge of insufficiently treated sewage from marine outfalls (i.e., pipe discharge) as a point-source (*Risk et al., 2009b*; *Al-Farawati, 2010*) was already shown to raise near-shore N availability (*Sawall et al., 2014b*), affect planktonic (*Pearman et al., 2018*) and coral (*Ziegler et al., 2016*) microbial communities and reach nearby reefs (*Risk et al., 2009b*; *Peña-García et al., 2014*). As nutrients, among several stressors, have the largest effect on Red Sea hard coral resilience to climate change (*Hall et al., 2018*), a deeper understanding of the community response to eutrophication is fundamental.

Benthic coral reef communities are crucial for many ecosystem functions, including the cycling and retention of carbon (C) and nitrogen (N) (*Johnson et al., 1995*; *Wild et al., 2004a*; *O'Neil & Capone, 2008*) but suffer from anthropogenic disturbances (*Hoegh-Guldberg et al., 2007*; *Carpenter et al., 2008*; *Hughes et al., 2018*). N availability is an important limiting factor for the biological productivity in oligotrophic reef environments (*Lesser et al., 2007*). Local eutrophication may impact reef organisms that typically grow in nutrient-poor waters (*Naumann et al., 2015*), and the diverse array of metabolisms they are comprised of. One prominent example is the entirety of coral host, endosymbiotic algae (zooxanthellae), bacteria and other microorganisms (*Wegley et al., 2007*), called the coral holobiont. The enrichment source (*Shantz & Burkepile, 2014*; *Burkepile et al., 2019*) and ratio of supplied nutrients is important to determine reef biota's reactions to eutrophication, particularly for corals (*Haas, Al-Zibdah & Wild, 2009*; *Wiedenmann et al., 2013*). Metabolic differences, for example between autotrophic and heterotrophic lifestyles, as well as the feeding environment of heterotrophic organisms, can lead to imbalances of essential biochemicals, which may become limiting (*Müller-Navarra, 2008*). Critical parameters to evaluate and trace nutrient fluxes as well as limitations in marine environments are the C and N elemental (*Goldman, 1986*; *Hillebrand & Sommer, 1999*; *Sterner & Elser, 2002*; *Jessen et al., 2013a*; *Stuhldreier et al., 2015*) and isotopic (*Risk et al., 2009a*; *Baker et al., 2010*; *Kürten et al., 2014*) composition. N uptake and circulation in the

reef might be fast and while the input of N can be measured by the long-term increase in forms of N concentrations (*Lapointe et al., 2019*), it is most directly traceable in the short-term by the isotopic signature of reef biota. As external sources and processes of N acquisition affect the isotopic composition, for example, of corals (*Hoegh-Guldberg et al., 2004*), anthropogenic N sources can be traced in the field (*Costanzo et al., 2001*; *Kendall, Elliott & Wankel, 2007*; *Baker et al., 2010*). N enrichment has negative effects on coral growth (*Ferrier-Pagès et al., 2000*; *Koop et al., 2001*; *Hall et al., 2018*), calcification (*Kinsey & Davies, 1979*; *Silbiger et al., 2018*), reproductive success (*Koop et al., 2001*; *Harrison & Ward, 2001*; *Loya et al., 2004*), biodiversity (*Duprey, Yasuhara & Baker, 2016*), bacterial communities (*Hall et al., 2018*) and increases the susceptibility of corals to bleaching (*Wooldridge & Done, 2009*; *Wiedenmann et al., 2013*; *Vega Thurber et al., 2014*; *Burkepile et al., 2019*). In contrast, other benthic groups in coral reefs, such as turf- and macroalgae benefit from increased nutrient availability in many cases (*Lapointe, 1987*; *Williams & Carpenter, 1988*), particularly in combination with reduced herbivory. Hence, shifts from coral- to algal-dominated reefs, so-called phase shifts, can occur (*Lapointe, 1997*; *Smith, Hunter & Smith, 2010*).

While extensive research investigated the causes of phase shifts (*McManus & Polsenberg, 2004*; *Norström et al., 2009*), nutrient effects on the ecophysiology and elemental stoichiometry of reef functional groups are rarely assessed, overlooking connections between uptake to utilization. Responding to the growing nutrient inputs to the central Red Sea, an assessment of their effects on coral reef communities is needed in this originally nutrient poor region, particularly which functional groups and ecophysiological parameters may indicate early-stage effects. Reefs in the oligotrophic Red Sea can serve as a "natural laboratory" (*Berumen et al., 2013*, *2019*; *Pearman et al., 2017*), as anthropogenic nutrient inputs add on a comparably low baseline. However, most studies have been conducted in the laboratory rather than in-situ, with associated risks of experimental artifacts, oversimplification or overestimation (see *Roth et al., 2019*). Indeed, local boundary layers and contact zones are of major importance in terms of direct interaction, small scale flow regimes as well as accumulation and transfer of organic matter (*Barott & Rohwer, 2012*; *Roach et al., 2017*), which can hardly be simulated under controlled laboratory conditions. The few similar studies that exist were conducted in less oligotrophic seas (*Koop et al., 2001*; *Den Haan et al., 2016*), along the natural environmental gradient of the Red Sea (*Kürten et al., 2014*), focused on one individual benthic group only (*Loya et al., 2004*; *Jessen et al., 2013a*, *2013b*), or only investigated benthic cover or chlorophyl content, not considering other metabolic parameters (*Haas, Al-Zibdah & Wild, 2009*; *Naumann et al., 2015*).

Therefore, we assessed the responses of major benthic functional groups (hard corals (*Pocillopora* cf. *verrucosa*, that is, tissue and zooxanthellae), soft corals (Xeniidae), turf algae and sediments) to N enrichment through a manipulative in-situ experiment in the central Red Sea. Combining elemental and stable isotope analysis, this approach provides information starting from N in the water column, through N uptake, to its utilization. We address the following underlying research questions: (1) Which major functional groups take up available N and (2) how did this affect organic carbon ($C_{org}$) and N
Table 1 **Relative benthic cover of functional groups at the experimental reef.** Data taken from *Roth et al. (2018)*.

| Major functional groups | Cover (%) |
|---|---|
| Filamentous turf algae | 36.8 |
| Hard coral | 28.8 |
| Rubble | 10.2 |
| Biogenic rock | 8.7 |
| Soft coral | 8.5 |
| Sediment | 6.0 |
| *Tridacna* sp. | 0.7 |
| Macroalgae | 0.4 |

contents? Taken together, we aimed to draw conclusions about nutrient limitation for different functional groups.

## MATERIALS AND METHODS

### Study site and environmental conditions

The experiments were conducted at Abu Shoosha reef (22°18′15″N, 39°02′56″E) on the west coast of Saudi Arabia in the central Red Sea from late January until late March 2018. The reef assessed in this study does not fall under any legislative protection or special designation as a protected area. Under the auspices of KAUST (King Abdullah University of Science and Technology, Thuwal, Saudi Arabia), sailing permits to the reef were granted that included the collection of corals and other reef benthos. This reef is characterized by generally high levels of herbivory and small fluctuations in ambient dissolved inorganic nitrogen (DIN) concentration during this period (*Roth et al., 2018*). For example, in January to March of the previous year (i.e., 2017), sea water concentrations of ammonium ($NH_4^+$) ranged from 0.16 to 0.17 µM, nitrate ($NO_3^-$) from 0.25 to 0.40 µM, nitrite ($NO_2^-$) from 0.03 to 0.06 µM, phosphate ($PO_4^{3-}$) from 0.02 to 0.21 µM and the resulting DIN ($NO_3^- + NO_2 + NH_4^+$)/$PO_4^{3-}$ ratio from 2.9:1 to 20:1 (*Roth et al., 2018*, Table S4). Abu Shoosha reef features turf algae (37%) and hard corals (29%) as most abundant functional groups (Table 1).

Key environmental variables were monitored every 2–3 weeks at the sampling site, as described in a related study by *Roth et al. (2018)*. Briefly, water temperature was measured with continuous data loggers (Onset HOBO Water Temperature Pro v2 Data Logger—U22-001; accuracy: ±0.21 °C) and are given in 3-day means (72 h). For background measurements of dissolved $NO_3^-$, $NO_2^-$ and $PO_4^{3-}$, water samples were taken in triplicates at the study site at least 2 m away from any fertilizer source (see "Experimental Design and Sampling Strategy" for more details). Water samples were filtered on the boat (Isopore™ membrane filters, 0.2 µm GTTP) and stored dark and cool until they were frozen to −50 °C in the lab. Nutrient concentrations were determined with a continuous flow analyzer (AA3 HR, SEAL). The limits of quantification (LOQ) for $NO_3^-$, $NO_2^-$ and $PO_4^{3-}$ were 0.084 µmol $L^{-1}$, 0.011 µmol $L^{-1}$ and 0.043 µmol $L^{-1}$ respectively. Five mL

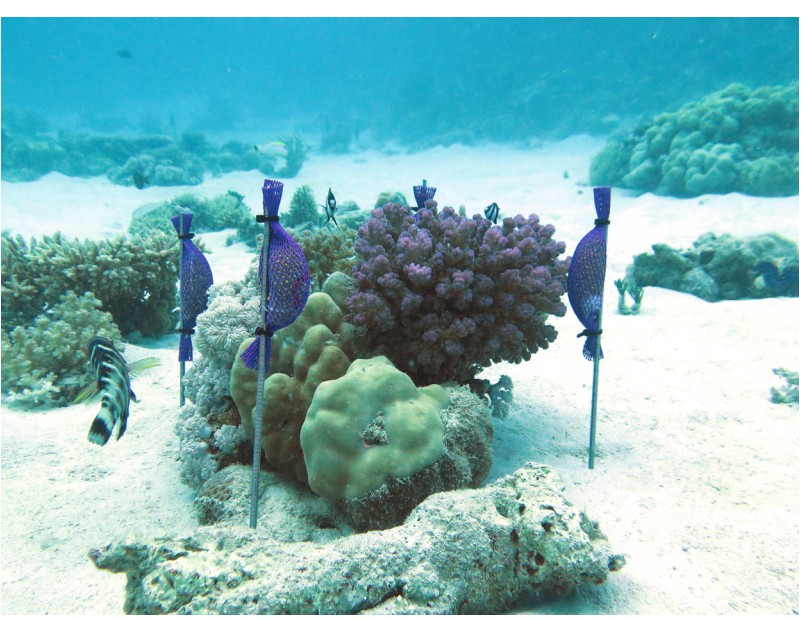

**Figure 1 Example of a manipulated in-situ community surrounded by four pins with attached fertilizer (Osmocote®) bags, photo: Florian Roth.**

subsamples for $NH_4^+$ were filtered into separate acid washed centrifuge tubes. A total of 1.2 mL ortho-phthalaldidehydesolution (OPA) was added, and samples were incubated >4 h with OPA in the dark. $NH_4^+$ concentrations were determined fluorometrically within 8 h (Trilogy® Laboratory Fluorometer; Turner Designs Inc., San Jose, CA, USA). The LOQ for $NH_4^+$ was 0.094 µmol L$^{-1}$. The sum of $NO_3^-$, $NO_2^-$ and $NH_4^+$ concentrations reflect DIN.

## Experimental design and sampling strategy

Eight distinct patches of reef communities, each surrounded by patches of reef sand, were chosen in the back reef of Abu Shoosha at a water depth of approximately 5 m. The chosen communities, which represented the surrounding reef in composition (Table 1), were exposed to simulated eutrophication for 8 weeks in total. More specifically, each of the replicate communities was surrounded by four pins with approximately 70 g of slow release fertilizer granulate (Osmocote® Plus (15-9-12)) (Fig. 1). Being one of the most commonly used fertilizers for eutrophication experiments (*Wheeler, 2003*; *Russell et al., 2009*; *Falkenberg, Russell & Connell, 2013*; *Stuhldreier et al., 2015*), this approach provides a fast and high supply of macronutrients (15% total N (8% nitrate N, 7% ammoniacal N), 9% available phosphate, 12% soluble potash) from the 1st day of fertilization under local temperature regimes (*Adams, Frantz & Bugbee, 2013*). Osmocote® Plus (15-9-12) provides a balanced fertilization of N and phosphorus (P), however, only the fate of N was considered in this experiment as particularly N effects were of interest. The fertilizer was renewed every 2–3 weeks to assure a continuous nutrient supply (*Adams, Frantz & Bugbee, 2013*). To test whether the nutrient addition was effective

locally, water samples for nutrients were taken directly at the fertilizer pin, and 25 cm towards the manipulated communities according to the protocol outlined above.

The effect of eutrophication was then assessed at the major functional groups (in terms of benthic reef cover in the central Red Sea) that were present in the selected communities. Specifically, we chose autotrophic hard corals (*Pocillopora* cf. *verrucosa)*, soft corals (Xeniidae), turf algae and reef sand (sediments). These groups covered ~80% of the sampled reef (*Roth et al., 2018*). Turf algae were defined as dense and flat (less than 2 cm in height) assemblages of filamentous algae of different species, including small individuals of macroalgae and cyanobacteria.

Manipulated specimens ("treatment") were sampled from within a close radius (~25 cm) of the fertilizer tubes. As the in-situ communities were also needed in other experiments investigating their C chemistry (F. Roth, 2018, unpublished data) and N fluxes (Y. El-Khaled, 2018, unpublished data), specimens for natural conditions at the beginning ("start") of the experiment were collected in the surrounding reef. Additional samples in replicates of eight were collected in the surrounding non-fertilized reef at the end ("control") of the experiment, to reflect non-fertilized control conditions. For start and control data, a distance of at least 10 m from any fertilizer pin was maintained and the same depth as well as light conditions were given.

Samples were acquired with hammer and chisel. Hard coral and turf fragments (their substrate) were of approx. 10 cm in length. Sediments were collected using a Petri dish, which was dragged into the sediment upside-down (max. depth 14 mm) and the sediment was fixed to the dish from underneath. The samples were stored at −80 °C until further preparation. Hard corals, soft corals and turf algae were rinsed with Milli-Q to remove excess salt. Epilithic turf algae were scraped off from their surface with a scalpel and tweezers.

## Elemental and stable isotopic compositions of C and N

Turf algae, soft corals and sediments were dried for 48 h (sediments: 72 h) at 40 °C. Following *Jessen et al. (2013b)*, hard coral tissue was removed using an airbrush, and the resulting tissue slurry was weighed, homogenized (MicroDisTec 125) and centrifuged for 10 min (Eppendorf Centrifuge 5,430 R, 4 °C, 3,220 rcf) to separate algae ("zooxanthellae") from animal "tissue". The supernatant was filtered (Whatman, GF/F) and for each sample two filters were generated. Filters for N and inorganic C analysis were dried for 24 h at 40 °C. Filters for $C_{org}$ measurements were exposed to HCl fumes (from 37% HCl) and dried for 24 h. The remaining zooxanthellae pellet was dried for 48 h at 40 °C.

Sub-samples of all groups were ground using an agate mortar and pestle. A mill grinder (Retsch, PM 200, 4 min) was used for the sediments. For preparation of $C_{org}$ samples, 5–10 g of ground sediment were placed in an Erlenmeyer flask and covered with Milli-Q. Drops of HCl (37%) were added until the reaction ceased. The acidified liquid was transferred to 50 mL Falcon tubes which were filled up with Milli-Q, to stepwise wash the sample pellet and raise the pH up to neutrality, and subsequently centrifuged for 10 min

(Eppendorf Centrifuge 5,430 R, 4 °C, 7,200 rcf). The liquid supernatant was discarded and tubes were then refilled with Milli-Q for 3–4 times to raise pH. The pellets were dried in the Falcon tubes for 48 h at 40 °C.

The dry, homogenous powder was analyzed for: (a) N and inorganic C quantities; and (b) $C_{org}$ as in *Roth et al. (2018)*. $C_{org}$/N ratios, fractions of organic and inorganic C and isotope ratios were measured as in *Rix et al. (2018)* using a Flash 1112 EA coupled to a Delta V IRMS via a ConfloIV-interface (Thermo Scientific, Waltham, MA, USA). Isotopic ratios are shown as $\delta^{13}C$ or $\delta^{15}N$ (‰) = $(R_{sample}/R_{ref} - 1) \times 1,000$. There, $R$ is the ratio of heavier:lighter isotope ($^{13}C$:$^{12}C$ or $^{15}N$:$^{14}N$). As reference, Vienna Pee Dee Belemnite was used for C ($R_{ref} = 0.01118$) and atmospheric nitrogen for N ($R_{ref} = 0.00368$).

## Zooxanthellae cell density and mitotic index

For hard corals, zooxanthellae cell density and the mitotic index were analyzed following the described sampling strategy (start, control and treatment), whereby "start" and "control" were from the surrounding reef. Aliquots of 20 µL homogenized tissue sample and 80 µL Milli-Q were vortexed (Gilson, GVLab) directly before taking 10 µL on an improved Neubauer Levy hemocytometer (0.0100 mm deep). Pictures were taken with a ZEISS Primovert microscope via Labscope (Version 2.5) from the 5 × 5 grid in 40-fold and randomly in 20-fold magnification. Manual counts of zooxanthellae and the mitotic index were related to the total amount of airbrushed slurry per individual. Here, clumps and inhomogeneous patches were not considered. For normalization to the coral surface area, 3D models for all coral skeletons were generated using the software AutodeskReCap Photo (v18.2.0.8).

## Data analysis

Nitrogen uptake by functional groups was assessed using stable isotope analysis. N utilization was assessed by elemental analysis, and $C_{org}$/N ratios served to identify nutrient limitations (*Lapointe, Littler & Littler, 1992*; *Hillebrand & Sommer, 1999*; *Sterner & Elser, 2002*; *Lapointe et al., 2005*), along with zooxanthellae cell density and mitotic index (for hard corals). Statistical analysis was conducted with RStudio (*R Core Team, 2017*). Xeniidae and *Pocillopora* cf. *verrucosa* were not abundant in all eight communities (only in 5 and 6, respectively). Due to logistical constraints, "start" data of soft corals was not available. A two-way ANOVA (factors: treatment, dominant functional group) showed no significant effect of community composition on our response parameters under N eutrophication, therefore data from more coral and more algae dominated communities were pooled. Significant differences between "start" and "control" as well as between "control" and "treatment" were checked with two-sample t-tests (test statistic: t) if test assumptions were fulfilled, otherwise Mann–Whitney–Wilcoxon Tests (test statistic: W) were applied. A similar approach was conducted for cell density of zooxanthellae and mitotic index per treatment. Tissue homogenization of the "start" samples was visually much worse than for "treatment" and "control" samples, but is shown for completeness and homogeneity.

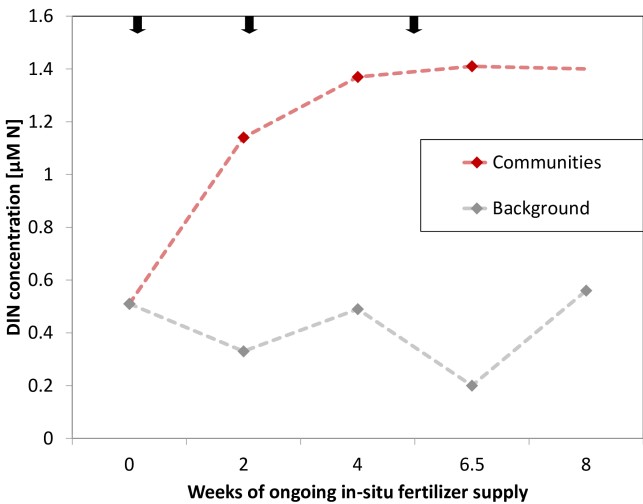

**Figure 2 Dissolved inorganic nitrogen (DIN) concentrations at experimental communities (red, last data point extrapolated) and of background sea water (grey) over time of in-situ manipulation.** Black arrows symbolize introduction and renewal of fertilizer.

## RESULTS

### Environmental parameters and N availability

During the study period, water temperature increased from 25 °C to 28 °C (Table S1). The mean background concentration in sea water for DIN was 0.34 ± 0.07 μM N and 0.10 ± 0.02 μM $PO_4^{3-}$ during the time of the experiment (measured after 2, 4 and 6.5 weeks). Accordingly, the environmental background DIN/$PO_4^{3-}$ ratio was 3.4 (±0.08):1 on average. The manipulation of nutrients increased DIN concentrations (measured after 2, 4 and 6.5 weeks) on average 3-fold and to a maximum of 7-fold directly at the communities relative to background concentrations (Fig. 2). Namely, manipulated $NO_3^-$ was 1.05 ± 0.09 μM and manipulated $NH_4^+$ was 0.22 ± 0.06 μM. $PO_4^{3-}$ remained at ambient condition, despite being present in the fertilizer (Table S1). As such, the mean DIN/$PO_4^{3-}$ ratio at the manipulated communities was 15.1 (±3.46):1.

### Uptake of excess N by benthic functional groups

Pure Osmocote® fertilizer was enriched in $^{15}$N ($\delta^{15}$N = 16.326 ± 0.257, Table S2). Hard corals (tissue), turf algae and soft corals took up excess N, as indicated by significantly ($t_{12}$ = 2.553, $p$ = 0.025; $t_{13}$ = 3.228, $p$ = 0.007; $t_9$ = 6.705, $p$ < 0.001, respectively) increased $\delta^{15}$N (Fig. 3A). The $\delta^{15}$N values in manipulated functional groups were 8% (*Pocillopora* tissue), 27% (Xeniidae) and 28% (turf algae) higher compared to untreated controls after the same time.

### Utilization of excess N by benthic functional groups

Nitrogen content was highest in hard coral zooxanthellae both before and after eutrophication (Fig. 3B). In the eutrophication treatment, N content was significantly higher in the tissues of Xeniidae ($t_9$ = 5.667, $p$ < 0.001) and turf algae ($W$ = 49, $p$ = 0.014).

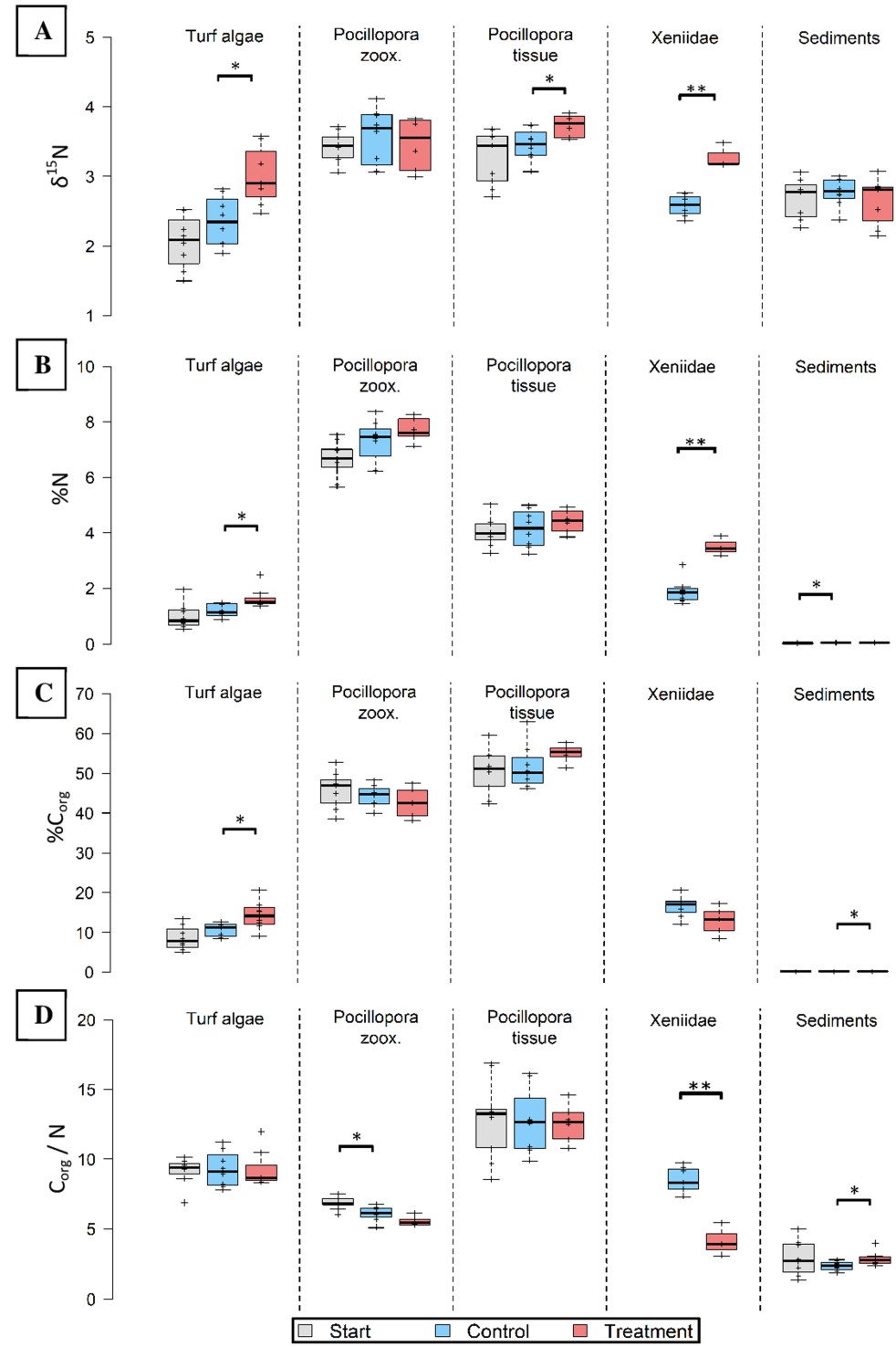

**Figure 3 Nitrogen (N) and carbon elemental and N isotopic composition of major functional groups before (grey), without (blue) and after 8 weeks in-situ eutrophication (red).** Investigated groups are turf algae, *Pocillopora* cf. *verrucosa* zooxanthellae ("zoox.") and -tissue, Xeniidae and sediments. Eight replicates per boxplot. (A) Nitrogen isotopes ($\delta^{15}N$), (B) nitrogen content (%N), (C) organic carbon content (%$C_{org}$), (D) organic carbon to nitrogen ratio ($C_{org}/N$). Asterisks indicate significant differences (*$p < 0.05$ and **$p < 0.001$).

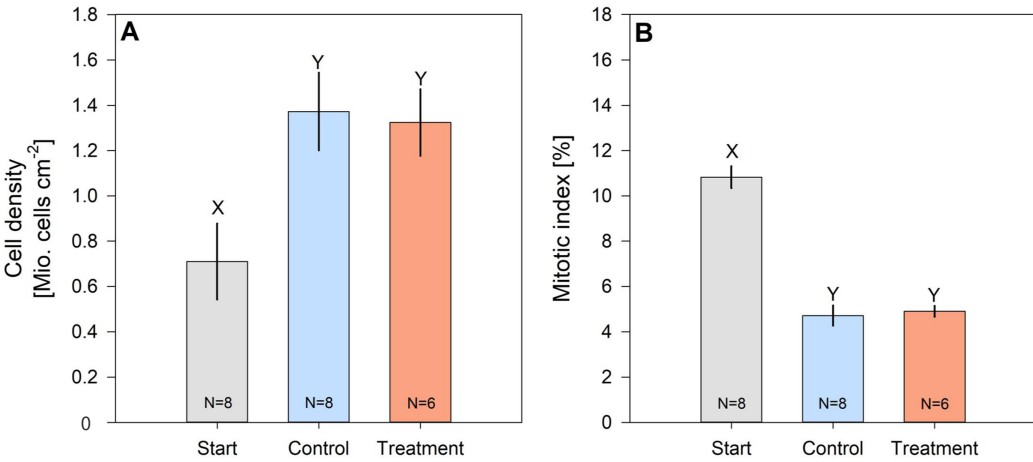

**Figure 4 Cell density (A) and mitotic index (B) of hard coral's (*Pocillopora* cf. *verrucosa*) zooxanthellae before (grey), without (blue) and with 8 weeks of N eutrophication (red).** Error bars represent the standard error of the mean, letters *X* and *Y* indicate significant differences.

Their tissues contained 85% (Xeniidae) and 39% (turf) more N compared to untreated controls. Increases in the other groups were not significant.

To investigate whether N was utilized to produce $C_{org}$ under a metabolically stable ratio, the $C_{org}$ content and $C_{org}/N$ ratio are presented. The hard coral components showed the highest $C_{org}$ content ranging from 50.39 ± 1.83% to 55.05 ± 1.49% in the tissue and between 42.64 ± 1.39% and 45.78 ± 1.49% in zooxanthellae (Fig. 3C). Minimum $C_{org}$ content was observed in reef sediments, ranging between 0.09 ± 0.01% and 0.11 ± 0.01%. Only turf algae and sediments showed a significant change in $C_{org}$ content ($t_{14} = 2.568$, $p = 0.022$; $t_{14} = 2.537$, $p = 0.023$, respectively). This represents an increase in $C_{org}$ content by 33% in turf algae and 24% in sediments in the treatment compared control specimen.

The $C_{org}/N$ ratio for treated Xeniidae was significantly lower ($t_8 = -6.405$, $p < 0.001$) than for Xeniidae in the surrounding reef (8.520 ± 0.320 compared to 4.132 ± 0.566, Fig. 3D). In sediments of the eutrophication "treatment" the $C_{org}/N$ ratio was significantly higher than in controls ($W = 53$, $p = 0.028$), however it did not increase compared to start values. In hard coral zooxanthellae, $C_{org}/N$ ratio declined over time but was not significantly different in treatment data compared to controls. $C_{org}/N$ remained constant in hard coral tissue and turf algae.

Cell density of hard coral zooxanthellae (*Pocillopora* cf. *verrucosa*) doubled over the 8 weeks, while their mitotic index halved (Figs. 4A and 4B). However, zooxanthellae density and mitotic index in fertilized and control corals remained similar. After 8 weeks, cell densities ranged from 1.324 ± 0.147 Mio. cells cm$^{-2}$ (treatment) to 1.373 ± 0.172 Mio. cells cm$^{-2}$ (control), whereas the mitotic index ranged from 4.718 ± 0.445% to 4.901 ± 0.244% in organisms under control and N enrichment conditions, respectively.

## DISCUSSION

Studies on the effects of eutrophication in the oligotrophic central Red Sea are scarce, and have, so far, focused on eutrophication effects on single functional groups only, used

natural gradients or left out impacts on the elemental stoichiometry. In a comparative in-situ approach we therefore provide an assessment of eutrophication effects on several major functional groups' ecophysiology using elemental and stable isotope analysis, drawing conclusions on N uptake and utilization.

## Uptake of excess N by major benthic functional groups

In-situ N enrichment resulted in an uptake of N in the tissues of turf algae, soft corals and hard corals, but not in sediments and hard coral zooxanthellae. The fact that turf algae exhibited the lowest $\delta^{15}N$ at the start of our experiment suggests considerable assimilation of N from $N_2$ fixation (*Yamamuro, Kayanne & Minagawao, 1995*; *Rix et al., 2015*; *Tilstra et al., 2017*; Y. El-Khaled, 2018, unpublished data). Biological fixation of atmospheric $N_2$ leads to a depletion in $^{15}N$ in the N compounds of the fixer (*Carpenter et al., 1997*; *Karl et al., 2002*). After the experiment, turf algae showed one of the strongest uptakes of N from the fertilizer among the benthic reef functional groups, as indicated by significantly higher $\delta^{15}N$ values (Fig. 3A), which concurs with *Den Haan et al. (2016)*. The low $DIN/PO_4^{3-}$ ratio in ambient waters at the reef further suggests a relatively low degree of P-limitation under ambient conditions, which may help to explain the strong uptake of N when available. Concordantly, *Lapointe et al. (2019)* showed that in the eutrophic waters of the Florida Keys, the N:P ratio of turf algae (293:1) increased to a much greater extent than that of macroalgae (71:1) as DIN concentrations increased over a 3-decade period. Also soft corals incorporated fertilizer N and reached higher $\delta^{15}N$ values than typical for soft corals that are exposed to industrial and urban run-off in the central Red Sea (*Kürten et al., 2014*).

In the present experiment, the uptake of excess N was not noticeable in the sediment $\delta^{15}N$, in contrast to a study by *Miyajima et al. (2001)*. There, sediment microflora took up $NO_3^-$ and $NH_4^+$ in bottle incubations (*Miyajima et al., 2001*), and assimilation as well as adsorption of N compounds on carbonate reef sands were observed (*Erler, Santos & Eyre, 2014*). This would suggest fast nutrient uptake, especially in microalgae on the sediments, and high uptakes into pore water, as reported by *Rasheed, Badran & Huettel (2003)* and *Erler, Santos & Eyre (2014)*. Pore water was not targeted in our study and a change of isotopic signature in the sediment could take longer than the current experimental period, as previously suggested by *Cook et al. (2007)* who did not find clear $\delta^{15}N$ patterns in N manipulated sediments. This would account for the integration time of isotopic signatures through the food-web (*Rolff, 2000*; *O'Reilly et al., 2002*). In agreement to our study, *Koop et al. (2001)* also did not find high $\delta^{15}N$ uptake in sediments. Potentially, organisms on and in the sediments are not N-limited, given that sediments are active sites of microbial N transformations (*Capone et al., 1992*) and remineralization (*Tribble, Sansone & Smith, 1990*) allowing for N recycling.

Within the hard coral holobiont, the zooxanthellae did not incorporate excess N significantly. This generally agrees with *Koop et al. (2001)* and *Den Haan et al. (2016)* showing that hard corals (i.e., *Madracis mirabilis* (now *Madracis myriaster*)) take up less excess nutrients than turf or macroalgae. Most studies, however, report stronger $\delta^{15}N$ enrichment in the zooxanthellae fraction compared to host tissue (*Grover et al., 2002*;

*Pernice et al., 2012*; *Kopp et al., 2013*). There are several possibilities why this was not observed in our study. Firstly, $NO_3^-$ uptake in zooxanthellae can be highest under low $NH_4^+$ availability (*Grover et al., 2003*; *Tanaka et al., 2017*), however, $NH_4^+$ was elevated ~5-fold compared to the environmental background in our experiment (Table S1). Secondly, P can be a limiting factor to zooxanthellae's N uptake (*Godinot et al., 2011*). Unlike the highly increased N availability, the P provided by our fertilizer did not alter the $PO_4^{3-}$ concentration 25 cm away from the source (Table S1). The resulting increased $DIN/PO_4^{3-}$ ratio at the communities underlines a stronger P-limitation under manipulation. Potentially, reef sediments (*Millero et al., 2001*) or organisms in the water column and the surrounding benthos took up $PO_4^{3-}$ too quickly as P was stated crucial (*Cuet et al., 2011*) and limiting (*Lapointe, Littler & Littler, 1992*; *Eyre, Glud & Patten, 2008*; *Kürten et al., 2014*) for primary production in coral reefs. However, we acknowledge dissimilar findings on the limiting roles of N and P in the central Red Sea (*Peña-García et al., 2014*). The understanding of P cycling and limitation in coral reef environments is still in its infancy (*Ferrier-Pagès et al., 2016*), but $PO_4^{3-}$ could have been limiting for significant N uptake in zooxanthellae (*Godinot et al., 2011*). In contrast to other findings (*Grover et al., 2003*; *Tanaka et al., 2006*), coral tissue incorporated more available N than the symbionts. This suggests that the host tissue was less P-limited than the zooxanthellate fraction, and hence took up relatively more N. This is corroborated by the low $DIN/PO_4^{3-}$ ratio of 3.4 in ambient waters of the studied reef that indicates N-rather than P-limitation, confirming *Al-Farawati, El Sayed & Rasul, (2019)*.

## Utilization of excess N by benthic functional groups

Due to eutrophication, tissue N content significantly increased in turf algae and soft corals but not in hard corals and sediments. $C_{org}$ content remained constant in hard and soft corals but increased in sediments and turf algae. Thus, turf algae and hard coral tissue remained at constant $C_{org}/N$ ratio, while it decreased in soft corals and showed unclear trends in hard coral zooxanthellae and sediments.

For turf algae, N and $C_{org}$ content were significantly higher under nutrient addition compared to controls, which contrasts findings by *Stuhldreier et al. (2015)* reporting no such eutrophication effects on turf algae dominated settlement communities. In the present study, relatively similar increases in N content (+39%) and $C_{org}$ content (+33%) occurred and the $C_{org}/N$ ratio stayed constant (between 9.1 and 9.3). Hence we interpret N to be a limiting nutrient (*Hecky, Campbell & Hendzel, 1993*) for turf algae growth, which also corroborates other studies (*Hatcher & Larkum, 1983*; *Williams & Carpenter, 1988*; *McCook, 1999*). Turf algae are strong opponents to corals (*Airoldi, 1998*; *Roth et al., 2018*), and their competitiveness under high-nutrient availability has been documented in Hawaii (*Smith, Smith & Hunter, 2001*), the Caribbean (*Vermeij et al., 2010*), Brazil (*Costa et al., 2000*), Australia (*Gorgula & Connell, 2004*) and in the Red Sea (*Naumann et al., 2015*). As turf algae are also rapidly taking over bare substrates (*Stuhldreier et al., 2015*; *Roth et al., 2018*) and are very resistant to disturbances (*Airoldi, 1998*), their monitoring should be on regional management agendas. Cover data was not documented in the present study but turf algae growth could be speculated upon based on increases in $C_{org}$ content.

Nitrogen was taken up by soft corals while the $C_{org}$ content did not increase, resulting in an altered elemental stoichiometry ($C_{org}$/N ratio). The strong decline in the soft corals' $C_{org}$/N ratio could be explained by an uptake of excess N as so-called "luxury consumption" (*Sterner & Elser, 2002*), describing on-going uptake while a different nutrient (e.g., P) might limit growth and productivity. We interpret that in our eutrophication experiment P rather than N was the limiting nutrient for soft corals, which may limit chlorophyl *a* content and photosynthesis in Xeniidae (*Bednarz et al., 2012*). Our data from elemental analysis and non-documented visual observations support the hypothesis of *Fabricius et al. (2005)* that soft corals could react more timely and strongly to water quality gradients than hard corals.

Our results further correspond to other studies (*Capone et al., 1992*; *Rasheed, Badran & Huettel, 2003*; *Wild et al., 2004b*) finding low $C_{org}$ content in carbonate dominated reef sands (0.18–0.36%), which were even lower in our study (0.1%). We acknowledge that the utilized acid wash-out processing may underestimate $C_{org}$ contents due to $C_{org}$ losses to the liquid acid of 4–52% (*Yamamoto, Kayanne & Yamamuro, 2001*) and as such lower the $C_{org}$/N ratio. However, this is a commonly used method in comparative studies (*Rasheed, Badran & Huettel, 2003*; *Wild et al., 2004b*). The observed 21% increase in $C_{org}$ could be attributed to P-supported algae growth on the sediments (Fig. S1) as fertilizer N was not taken up (constant $\delta^{15}$N) but gross primary production significantly increased (Y. El-Khaled, 2018, unpublished data). A different source for the increased $C_{org}$ content in sediments could be the export of $C_{org}$ from turf algae (F. Roth, 2018, unpublished data) for example, as dissolved organic carbon (DOC) (*Haas et al., 2011*) and subsequent uptake by reef sediments, as suggested by *Cárdenas et al. (2015)*. This, along with the low $C_{org}$/N ratio, corroborates the previous assumption that life in and on the sediments, as well as its increase in $C_{org}$ content was not N limited.

Regarding the hard coral holobiont, our results suggest that the incorporation of excess N only to the host tissue did not result in its utilization in terms of $C_{org}$ production. As such, $C_{org}$/N ratios for *Pocillopora* cf. *verrucosa* contrast a study conducted in 10 km distance to our study site further offshore at the same time of the year in 2012 (*Ziegler et al., 2014*). There, the natural host total C:N ratio was around 5, which makes our presented host material appear more N depleted in comparison. Over time, we observed an increase in symbiont cell density, which contrasts with other studies finding higher zooxanthellae cell densities in *Pocillopora* species in cooler and more nutrient rich phases (*Stimson, 1997*; *Al-Sofyani & Floos, 2013*; *Sawall et al., 2014a*). However, particularly the similarity between treatment and non-fertilized controls should be considered where cell density and mitotic index did not differ. Similar findings have been reported by *Ferrier-Pagès et al. (2001)* and *Rosset et al. (2017)* during pure N fertilization but contrast with other studies (*Stambler et al., 1991*; *Muller-Parker, Cook & D'Elia, 1994*; *Fabricius, 2005*). Increased zooxanthellae cell density in hard corals was found, for example, after only 18 days of eutrophication (*Falkowski et al., 1993*) or following a natural nutrient gradient (*Sawall et al., 2011*). Altogether, this suggests that N was not a limiting factor for zooxanthellae in our experiment. The production (*Ezzat et al., 2016*), health and density (*Tanaka et al., 2017*) of zooxanthellae cells was found to be P limited. Accordingly, high

P availability resulted in higher increases of zooxanthellae density (*Pocillopora damicornis* and *Euphyllia paradivisa* (now *Fimbriaphyllia paradivisa*)) than availability of only N (*Stambler et al., 1991*; *Rosset et al., 2017*). As a consequence, we hypothesize that hard corals also did not shift in primary productivity, even though a significant increase in $\delta^{13}$C in hard coral zooxanthellae (Fig. S2) could be a sign of increased photosynthesis (*Swart, Saied & Lamb, 2005*) for example, following a seasonal pattern (F. Roth, 2018, unpublished data). However, gross primary production did not increase in our manipulated hard corals (Y. El-Khaled, 2018, unpublished data). The increased $\delta^{13}$C in the zooxanthellae (Fig. S2) could also be an indicator for a negative effect on hard coral health which was also found in relation to bleached *Favia favus* (now *Dipsastraea favus*) corals in the Northern Red Sea (*Grottoli, Tchernov & Winters, 2017*) but not *Montastraea faveolata* (now *Orbicella faveolata*) in Florida (*Wall et al., 2019*). Given our 8-week observation period and a comparatively cold water temperature, our study did not provide a setting to trace severe bleaching effects and for the Southern Red Sea it was speculated that higher nutrient availability might even benefit *P. verrucosa* to resist higher water temperature (*Sawall et al., 2014a*). This corroborates that effects of eutrophication on coral health are not always negative (*Bongiorni et al., 2003*; *Sawall et al., 2011*; *Ezzat et al., 2019*) and do not necessarily harm or kill individual coral colonies but get outcompeted or overgrown over time (reviewed in *Fabricius, 2005*). Longer (3 years; *Vega Thurber et al., 2014*), and both longer and stronger (1 year, 36.2 μM $NH_4^+$; *Koop et al., 2001*) N manipulation could, however, lead to increased coral mortality (*Koop et al., 2001*). In particular, reviewed findings (*Morris et al., 2019*), natural long-term observations (*Lapointe et al., 2019*) and laboratory experiments (*Wiedenmann et al., 2013*; *Rosset et al., 2017*) with high N (>3 μM and 38 μM N, respectively) and low P supply (<0.07 μM and 0.18 μM P, respectively) increased susceptibility of corals to bleaching, which suggests negative effects. In agreement with *Ezzat et al. (2016)* and *Ferrier-Pagès et al. (2016)*, we suggest increasing efforts investigating P cycling and limitation in current and future reef ecosystems. Besides this key role of nutrient ratios, *Burkepile et al. (2019)* highlight the importance to also account for varying effects of different forms of N. As N sources and pathways in corals and their reef environments are of major importance to better understand ecosystem functioning (*Rädecker et al., 2015*), the uptake and utilization of N (this study) should be compared to eutrophication effects on the N cycle.

## CONCLUSIONS

Anthropogenic pressures on the Red Sea are constantly increasing (*Carvalho et al., 2019*) and 60% of Red Sea coral reefs are at stake (*Burke et al., 2011*). We were able to show cascaded, group-specific responses to N availability and link elemental and isotopic composition to group-specific nutrient limitations, N uptake and utilization, and highlight the importance of P limitations in hard and soft corals. Even over an 8-week N eutrophication and under high abundance of herbivores, significant uptake and utilization of fertilizer N was shown particularly for turf algae as strong competitors for space in struggling reef ecosystems. As such, our study corroborates that turf algae can be early indicators for changes and anthropogenic influence (*Barott et al., 2012*; *Roth et al., 2015*),

reacting faster to eutrophication than hard coral zooxanthellae. As turf algae play a key role in phase shifts, are strong competitors to corals, rapidly take over bare substrates and are highly persistent, their substantial biochemical benefits from N supply should push coastal management to not only consider limiting future discharges but try to reduce both point-sources and non-point sources of nutrients already in place. Given the increasing coastal development in the central Red Sea, water quality management is challenged to improve future reef states (*Gurney et al., 2013*; *D'Angelo & Wiedenmann, 2014*) and should be on regional agendas for coastal urban development and aquaculture. The context in which eutrophication effects should be seen comprises further local (e.g., fishing pressure and habitat destruction) and global (e.g., warming and ocean acidification) factors to which coastal development adds high nutrient loads on top. Low N concentrations were shown to be a crucial precondition for coral recovery (*Robinson, Wilson & Graham, 2019*) and particularly in the Red Sea the maintaining of oligotrophic conditions could be the key factor and challenge for coral health and resilience to climate change (*Hall et al., 2018*).

## ACKNOWLEDGEMENTS

We are thankful to Marianne Falk for helping in elemental-and isotopic analysis and Nils Rädecker for coral sample processing advice. We were happy to get further support by Pedro Ruiz-Compean (sediment samples), Aislinn Dunne (sediment samples), Sophia Tobler (image editing), Rodrigo Villalobos and João Cúrdia (both field work).

### Funding

This research was supported by DFG grant Wi 2677/9-1 to Christian Wild, KAUST baseline funding to Burton H. Jones and the KAUST VSRP program to Denis B. Karcher. Susana Carvalho is funded by the Saudi Aramco-KAUST Center for Marine Environmental Observations. The funders had no role in study design, data collection and analysis, decision to publish, or preparation of the manuscript.

### Grant Disclosures

The following grant information was disclosed by the authors:
DFG: Wi 2677/9-1.
KAUST.
KAUST VSRP Program.
Saudi Aramco-KAUST Center for Marine Environmental Observations.

### Competing Interests

The authors declare that they have no competing interests.

### Author Contributions

- Denis B. Karcher conceived and designed the experiments, performed the experiments, analyzed the data, prepared figures and/or tables, authored or reviewed drafts of the paper, and approved the final draft.

- Florian Roth conceived and designed the experiments, performed the experiments, analyzed the data, authored or reviewed drafts of the paper, and approved the final draft.
- Susana Carvalho conceived and designed the experiments, authored or reviewed drafts of the paper, and approved the final draft.
- Yusuf C. El-Khaled performed the experiments, prepared figures and/or tables, authored or reviewed drafts of the paper, and approved the final draft.
- Arjen Tilstra performed the experiments, authored or reviewed drafts of the paper, and approved the final draft.
- Benjamin Kürten analyzed the data, authored or reviewed drafts of the paper, and approved the final draft.
- Ulrich Struck performed the experiments, authored or reviewed drafts of the paper, and approved the final draft.
- Burton H. Jones conceived and designed the experiments, authored or reviewed drafts of the paper, and approved the final draft.
- Christian Wild conceived and designed the experiments, analyzed the data, authored or reviewed drafts of the paper, and approved the final draft.

### Data Availability

The raw data are available in the Supplemental Files.

### Supplemental Information

Supplemental information for this article can be found online at http://dx.doi.org/10.7717/peerj.8737#supplemental-information.

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
