# Peer review of "Nitrogen eutrophication particularly promotes turf algae in coral reefs of the central Red Sea"

_PeerJ, doi:10.7717/peerj.8737_

## Round 0.1 · original submission · Minor Revisions

Two expert reviewers have evaluated your manuscript and their comments can be seen below. Both reviewers have very favourable comments, but they also have made some observations that should be taken into account..

Reviewer 1 ·

Basic reporting

.

Experimental design

.

Validity of the findings

.

Additional comments

Review of PeerJ 42898 - “Nitrogen eutrophication particularly promotes turf algae in coral reefs of the central Red Sea” by Karcher et al.

This paper provides exciting new data on how increasing human nitrogen enrichment affects coral reef communities in the Red Sea. The approach and methods are solid and the paper is well-written and referenced. I have made a few minor comments below, and suggestions for additional references and perspectives that could further strengthen the paper. I did not have access to the Figures or Tables, so cannot comment on those. Overall, well worthy of publication in PeerJ.

Abstract

Line 29, what about non-point source? Septic tanks, cesspits, atmospheric deposition?

Line 34 , what are the “background” N concentrations?

Line 35, what were the enriched NH4 and NO3 concentrations?

Line 42, did the N enrichment actually lead to increased productivity (growth) of turf algae or is that assumed based on higher Corg content?

Line 110, how about experimental artifacts?

Line 134, important to add here the mean values and range for ammonium, nitrate, and phosphate, as well as DIN/phosphate ratio, if possible, for the reef study site.

Line 151, spelling, fluorometrically

Introduction

Line 60, are these sewage discharges from point-source outfalls or from non-point source, such as septic tanks/cesspits?

Line 85, Revise this sentence to read “and while the input of N can be measured by the long-term increase in forms of N concentrations (Lapointe et al. 2019), it is most directly traceable in the short-term by the isotopic signature of reef biota.”

Line 159, replace “ca.” with approximately or “~”

Line 253, might want to add here that the DIN/PO4 ratio is 3.4:1. That could indicate a relatively low degree of PO4 stress (starvation) in reef corals. What was the major form of DIN in the fertilizer? Ammonium, nitrate, or both?

Line 310, could add here: “In the eutrophic waters of the Florida Keys, the N:P ratio of turf algae (293:1) increased to a much greater extent than that of macroalgae (71:1) as DIN concentrations increased over a 3-decade period (Lapointe et al. 2019).”

Line 347/348, the low DIN/PO4 ratios in ambient waters at the reef would suggest a relatively low degree of P-limitation, which helps explain the greater uptake of N than P.
Line 364, add Caribbean Sea and cite: Vermeij MJ, Van Moorselaar I, Engelhard S, Hörnlein C, Vonk SM, Visser PM (2010) The effects of nutrient enrichment and herbivore abundance on the ability of turf algae to overgrow coral in the Caribbean. PLoS One 5(12):e14312. https://doi.org/10.1371/ journal.pone.0014312
Line 456, add “both point-source and non-point source…” The non-point sources include septic tanks and cesspits that are major contributers to N enrichment in many urbanized coastal waters.

Reviewer 2 ·

Basic reporting

This work reports on experiments carried out in the Red Sea on the impacts of eutrophication on the nitrogen and carbon content and stable isotope composition of hard coral, soft corals, sediments and zooxanthellae. The authors do a very thorough job reporting on the background research and sometimes confounding results in the literature. The structure of the written work as well as tables and figures were easy to follow without being overly simplistic.

There were a couple grammatical errors / typos that can be easily fixed:

line 97: Smith et al. 2010
line: 242: "however also showed". this was confusing
line 350: use of "remained" was confusing

Experimental design

The experimental design was robust and all the idiosyncrasies in sample numbers were explained by the author and were the result on conducting experimental work in situ. The statistical analysis of the work was also robust and used appropriate techniques based on the data.

The techniques used to measure the impact of localized eutrophication were multidimensional and thus added depth to the overall interpretation of the results.

Validity of the findings

The finding of the research are largely consistent with previous work and where it differs the authors offer likely explanations. The discussion and conclusion are well developed and succinctly relate their work to that of the larger body of knowledge in this field.

---

## Round 0.2 · accepted · Accept

I am satisfied with the changes made to the manuscript.